# Optimization of Processing Conditions and Mechanical Properties for PEEK/PEI Multilayered Blends

**DOI:** 10.3390/polym14214597

**Published:** 2022-10-29

**Authors:** Sebastián Andrés Toro, Alvaro Ridruejo, Carlos González, Miguel A. Monclús, Juan P. Fernández-Blázquez

**Affiliations:** 1Department of Materials Science, Universidad Politécnica de Madrid, ETSI Caminos, C/Profesor Aranguren, 3, 28040 Madrid, Spain; 2Departamento de Ingeniería Mecánica, Universidad de Santiago de Chile, USACH, Av. Bernardo O’Higgins 3363, Santiago de Chile 9170022, Chile; 3IMDEA Materials Institute, C/Eric Kandel 2, Getafe, 28906 Madrid, Spain

**Keywords:** multilayer polymers (MLPs), polyetheretherketone (PEEK), polyetherimide (PEI), interfacial adhesion, interfacial diffusion, heterogeneous blends, thermoplastic composites, mechanical properties, thermoplastic processing

## Abstract

The goal of producing polyetheretherketone/polyetherimide (PEEK/PEI) blends is to combine the outstanding properties that both polymers present separately. Despite being miscible polymers, it is possible to achieve PEEK/PEI multilayered blends in which PEEK crystallinity is not significantly inhibited, as opposed to conventional extruding processes that lead to homogeneous mixtures with total polymer chain interpenetration. This study investigated a 50/50 (volume fraction) PEEK/PEI multilayered polymer blend in which manufacturing parameters were tailored to simultaneously achieve PEEK–PEI adhesion while keeping PEEK crystallinity in order to optimize the mechanical properties of this heterogeneous polymer blend. The interface adhesion was characterized with the use of three-point bending tests, which proved that a processing temperature below the melting point of PEEK produced weak PEEK–PEI interfaces. Results from differential scanning calorimetry (DSC), dynamic mechanical analysis (DMA), and X-ray diffraction analysis (XRD) showed that under a 350 ${}^{{}^{\circ}}$C consolidation temperature, there is a partial mixing of PEEK and PEI layers in the interface that provides good adhesion. The thickness of the mixed homogeneous region at this temperature exhibits reduced sensitivity to processing time, which ensures that both polymers essentially remain separate phases. This also entails that multilayered blends with good mechanical properties can be reliably produced with short manufacturing cycles. The combination of mechanical performance and potential joining capability supports their use in a wide range of applications in the automotive, marine, and aerospace industries.

## 1. Introduction

Thermoplastic materials have attracted attention in engineering applications due to their specific strength, high-impact resistance, shorter processing times, and reprocessing capability as compared to conventional thermosets [1,2,3,4].

In this regard, polyetheretherketone and polyetherimide are two high-performance thermoplastics that are increasingly being applied in the automotive and aerospace industries [5,6,7,8]. PEEK is a semicrystalline, aromatic thermoplastic with high chemical resistance, good tribological response, and excellent tensile fatigue behavior [9], and it is a suitable replacement for metal in a wide variety of high-temperature applications. PEI is an amorphous thermoplastic polymer with a glass transition temperature of around 217 ${}^{{}^{\circ}}$C, which results in remarkable thermo-oxidative stability and allows it to present good mechanical properties even at high temperatures [10,11]. PEI consists of ether groups that improve its melt processability without spoiling melt stability [12].

Taking into account the comparative advantages of each thermoplastic (e.g., the thermal stability of PEI and the chemical resistance of PEEK), it is of great interest to combine their properties and minimize their respective drawbacks by means of a PEEK/PEI blend, whose molecular miscibility is a proven fact [13,14,15,16]. Until now, the melt blending process, which commonly uses a twin-screw extruder, has been the most widely used technique in manufacturing PEEK/PEI polymer blends [17,18,19,20,21,22]. Although full PEEK/PEI interpenetration has been shown to provide good thermal and mechanical properties, there are significant disadvantages in terms of the physical and chemical properties of these materials when both are found in a homogeneous mixture. PEEK is a semicrystalline polymer, while PEI is an amorphous one. This implies that after total integration and subsequent cooling, PEI will remain unchanged [23], but the diffusion and infiltration of PEI chains into PEEK will inhibit the nucleation and growth of crystallites in the latter polymer. Reduced crystallinity has direct consequences on both the mechanical properties of the homogeneous polymer blend [24,25] and its ability to form a good bonding interface in PEEK-reinforced composite materials [26,27]. The multilayer polymer (MLP) stacking technique is an alternative strategy to handle these drawbacks. The MLP concept was originally established in 1966 through the layer-by-layer assembly technique (LbL) [28], and in recent years, it has been used in a wide number of applications, such as food packaging, drug delivery, catalysts, oil–water separation, and in aerospace applications [29,30,31,32]. Depending on the processing conditions, MLP can lead to homogeneous blends, or, as in composite materials, polymer layers can be kept as separate phases. In a previous work [33], it was shown that this strategy allowed for the crystallinity of PEEK to be retained when a processing temperature of 380 ${}^{{}^{\circ}}$C was used to obtain PEEK/PEI MLPs. At this temperature, there are two limitations: Firstly, PEI has a low viscosity and therefore easily undergoes excessive bleeding. Secondly, as processing time increases, PEI can penetrate further into PEEK, causing an additional homogenization of the blend and hindering PEEK crystallization.

In this work, PEEK/PEI multilayers with a 50/50 volume ratio were manufactured under different consolidation times and temperatures to determine the parameters that maximize the adhesion of both polymers and minimize the loss of crystallinity for PEEK. To this end, the fabricated MLPs were initially subjected to three-point bending tests to assess their resistance to delamination. After this, the thermal properties and microstructural heterogeneity of the blends were studied with the use of DSC, DMA, and XRD measurements. Nanoindentation was employed to analyze the change of mechanical properties along the multilayer cross-section. Lastly, tensile tests were used to characterize the mechanical properties of neat polymers and PEEK/PEI blends. The novelty of this work lies in its experimental confirmation that a certain set of processing conditions allows the multilayered configuration of two high-performance thermoplastic polymers to keep its original heterogeneity after manufacturing, regardless (to a great extent) of processing time. In particular, processing temperatures slightly above the melting point of PEEK, even for a period as short as 10 min, led to a robust PEEK–PEI interface while preserving PEEK crystallinity. This can greatly facilitate the fabrication, joining, or repair of components and opens the way to other technological alternatives, such as the processing of PEEK/PEI multilayers by means of additive manufacturing, which remain largely unexplored.

## 2. Materials and Methods

### 2.1. Samples Preparation

PEEK (Dexnyl© PEEK-SF) and PEI (Dexnyl© PEI-SF) were supplied by Bieglo GmbH, Hamburg, Germany in 500 μm thick films, and a volume blend ratio of 50/50 was used in this work. Multilayer materials were processed by alternately stacking three PEEK films and three PEI films (6 layers in total). These multilayers were subsequently consolidated through thermoforming processes (see Figure 1b) on a hot-press plate machine (LabPro 400, Fontijne Presses, Delft, The Netherlands) with temperature and force control. Using an aluminum mold, laminates of 280 × 280 mm${}^{2}$ were obtained with the application of three different consolidation schemes: 330 ${}^{{}^{\circ}}$C for 120 min (PEEK/PEI@T330t120), 350 ${}^{{}^{\circ}}$C for 10 min (PEEK/PEI@T350t10), and 350 ${}^{{}^{\circ}}$C for 120 min (PEEK/PEI@T350t120). The choice of these temperatures was made due to the fact that PEEK melts at approximately 340 ${}^{{}^{\circ}}$C. Therefore, studying the adhesion of this material with PEI at temperatures of ±10 ${}^{{}^{\circ}}$C relative to its melting point was of interest. In this sense, it was also crucial to consider the processing setup’s limitations: the hot press can achieve reasonable temperature control over the full multilayer, with a precision of a few Celsius if processing times are long, but less accurate if they are shorter. Therefore, in order to obtain precise results, the size step temperature of 10 ${}^{{}^{\circ}}$C seemed reasonable.

In all cases, a heating rate equal to 10 ${}^{{}^{\circ}}$C/min was set until the dwell temperature was reached. From this point on, a force of 100 kN was applied. After reaching the end of dwell (remanence) time, cooling was prescribed at a rate equal to 10 ${}^{{}^{\circ}}$C/min until room temperature was reached. Neat PEEK and PEI laminates were also manufactured by stacking six films of each polymer and replicating the same conditions for PEEK/PEI@T350t10. The stacking sequences are shown in Figure 1c. During consolidation, there was some squeeze-out flow, and so the final thickness of specimens was slightly reduced, depending on the processing parameters, from 3.0 mm (nominal thickness, 6 × 500μm) to 2.78 mm (PEEK/PEI@T330t120), 2.75 mm (PEEK/PEI@T350t10), and 2.60 mm (PEEK/PEI@T350t120).

### 2.2. Experimental Techniques

#### 2.2.1. Three-Point Bending Tests

In order to assess the influence of the consolidation scheme on the adhesion of the PEEK/PEI interface, five samples of each material were subjected to three-point bending (3PB) tests, following the ASTM D790 standard [34]. An Instron 1122 universal testing machine (Norwood, MA, USA) fitted with a 1 kN load cell was used and operated at a cross head speed of 1 mm/min. The dimensions of the specimens were 130 mm × 13 mm × thickness (2.78, 2.75, or 2.60 mm, depending on the consolidation cycle). The distance between supports was equal to 45 mm. The laminates were not symmetric, but behavior was not affected by any bending after manufacturing as a result of different thermal expansion coefficients.

#### 2.2.2. Differential Scanning Calorimetry

Differential scanning calorimetry (DSC) was performed on a TA model Q200 device (TA Instruments, New Castle, DE, USA), and nitrogen was used as a purge gas. The thermal behavior of the films was obtained by heating from 40 ${}^{{}^{\circ}}$C to 400 ${}^{{}^{\circ}}$C, at a heating rate of 10 ${}^{{}^{\circ}}$C/min.

#### 2.2.3. Dynamic Mechanical Analysis

Dynamic mechanical analysis (DMA) tests were performed on a DMA Q800 (TA Instruments, New Castle, DE, USA) in single cantilever mode, from −140 ${}^{{}^{\circ}}$C to 250 ${}^{{}^{\circ}}$C and at a rate of 2 ${}^{{}^{\circ}}$C min${}^{-1}$, with a constant frequency of 5 Hz. Rectangular samples with dimensions of 30 × 10 × 3 mm${}^{3}$ were used.

#### 2.2.4. X-ray Diffraction (XRD)

XRD patterns were recorded on the neat (as-received) and multilayer polymer samples using in an Empyrean X-Ray difractometer (Malvern Panalytical Ltd., Malvern, UK) with Ni-Altered Cu Kα radiation (λ = 1.54 $\dot{A}$). The sample–detector distance was set at 10 cm, which allowed for XRD data in the range of 2.5–3.5 cm. XRD data in the 2θ range was from approximately 5${}^{{}^{\circ}}$ to 45${}^{{}^{\circ}}$. The X-ray generator power was set at 40 kV and 30 mA. The scanning speed used was 1${}^{{}^{\circ}}$/min, and the sampling width was 0.05${}^{{}^{\circ}}$.

#### 2.2.5. Tensile Tests

Five samples of each material (neat PEEK, neat PEI, PEEK/PEI@350t10, and PEEK/ PEI@350t120) were tested at ambient temperature using an Instron 8501 servohydraulic universal test machine with a loading capacity of 10 kN, following the requirements of ASTM D638 [35], manufacturing type V tensile test samples. For load measurement, a cross-head speed of 1 mm/min was used.

Full displacement and strain fields during the uniaxial tensile test were measured with a commercial digital image correlation code (VIC 2D, Correlated Solutions Inc., Irmo, SC, USA) [36] by post-processing the images extracted from a Nikon D5600 camera (magnification of 4× and resolution of 16.9 megapixel) at regular intervals. For image correlation, the speckle pattern on the surface of the samples was generated using water slide paper [37].

#### 2.2.6. Nanoindentation Tests

Nanoindentation was performed with a NanoTest instrument from Micro Materials Ltd. (Wrexham, UK), equipped with a diamond Berkovich tip, whose area function was previously calibrated using a standard fused silica sample. The load–displacement curves were recorded using a maximum load of 70 mN, loading at a constant strain rate of 0.05 s${}^{-1}$. In all cases, a dwell time at a peak load of 20 s was used, followed by a final unloading for 2 s. In the current work, the Oliver–Pharr analysis was used to determine the reduced elastic modulus ($E_{r}$) of the samples calculated from the load–displacement curves, as expressed in Equation (Equation 1) [38]: (1)$$E_{r}=\beta \frac{S\sqrt{\pi }}{2A},$$
where the term β is a constant related to indenter geometry (β = 1.034 for a Berkovich indenter), *S* corresponds to the elastic unload stiffness, and *A* is the contact area. In this study, the reduced elastic modulus profiles along the thickness of the multilayer laminate were obtained using the grid nanoindentation methodology. The indents were arranged into 96 rows and 3 columns, separated by a distance of 2000 μm (C${}_{1}$, C${}_{2}$, and C${}_{3}$) and an even spacing of 25 μm between indents in each of the columns. As shown in Figure 2, the grid is contained within an indentation region of interest (ROI) that includes the four inner layers in order to avoid possible defects in the contour of the sample.

The representative reduced modulus profile for each sample was obtained by averaging the $E_{r}$ values of the three columns mentioned above.

## 3. Results and Discussion

### 3.1. PEEK/PEI Interface Integrity

Figure 3 shows three representative force–displacement curves for each material recorded during the three-point bending tests. It can be clearly observed that the PEEK/PEI multilayers consolidated at 330 ${}^{{}^{\circ}}$C (i.e., under the melting point of PEEK) were not able to withstand significant loads before failing at forces well below the other two specimen groups. However, the force–displacement curves of the multilayers consolidated at 350 ${}^{{}^{\circ}}$C exhibited a much better response in terms of maximum load and deflection. In these cases, regardless of the processing time (10 or 120 min), no delamination was observed at the end of the test.

Regarding the specimen consolidated at 330 ${}^{{}^{\circ}}$C for 120 min (PEEK/PEI@330t120), the interlayer adhesion failure can be seen as the drop in the load–displacement curve (Figure 3a) and directly observed as a delamination crack between two of the PEEK and PEI films (Figure 3b). This occurred in all PEEK/PEI@330t120 tested specimens and shows that despite the long consolidation time, a temperature of 330 ${}^{{}^{\circ}}$C is not enough to obtain full adhesion between PEEK and PEI. For this reason, from this point onwards, only PEEK/PEI@T350t10 and PEEK/PEI@T350t120 multilayers will be considered.

### 3.2. Thermo-Mechanical Properties

The excellent adhesion observed in specimens processed at 350 ${}^{{}^{\circ}}$C was due to the interpenetration of the polymer chains between the layers. This diffusion of polymer chains is possible above 340 ${}^{{}^{\circ}}$C, where both polymers are in the molten state. At 330 ${}^{{}^{\circ}}$C, although PEEK is over its glass transition, its semicrystalline state prevents the diffusion of polymer chains between layers, causing the failure of the interlayer. At 350 ${}^{{}^{\circ}}$C, two multilayers were processed under the same pressure and temperature conditions, but one was kept at dwell temperature for 10 min and the other for 120 min. DSC experiments were performed on both samples (Figure 4). No relevant differences were found; separate glass transitions for both polymers were observed, with just a tiny shift to a higher temperature for PEEK and to a lower temperature for PEI in the sample processed for 120 min. This was due to the miscibility in the interface, and this miscibility also affected the melting enthalpy, which was 24.3 J/mol for the 120 min processed sample and 26.9 J/g for the 10 min one. This reduction is due to the mixture of polymer chains that occurs at the interface, where PEI chains hinder the crystallization of PEEK. However, this effect is less pronounced at 350 ${}^{{}^{\circ}}$C in comparison to that observed at 380 ${}^{{}^{\circ}}$C in our previous research [33]. The part of the curve corresponding to the cooling ramp confirmed these observations, such as in the lower crystallization enthalpy (28.7 J/g for 120 min vs. 29.9 J/g s for 10 min) or in the small shifts observed in the glass transition. The loss of crystallinity in PEEK due to the presence of PEI was dramatically confirmed in the extruded blends, where there was a perfect miscibility between both polymers that suppressed the formation of PEEK crystallites. [18,39].

X-ray diffraction experiments were carried out by reflection. Therefore, two experiments were performed on each MLP sample: one on the outer PEEK face and the other on the outer PEI face (see Figure 5). No significant differences were found. As expected, the amorphous halo was observed as a broad peak in the outer PEI layer. In the case of the outer PEEK layer, there were no differences in the position and width of the peaks, and only a small reduction in the intensity of the peaks was observed for the samples processed at 120 min. Considering that the penetration of the X-ray under the conditions in which the experiments had been performed was around 500 μm, which is the pristine thickness of the films, we can conclude that the stability and isolation of the layers were not affected by the longer processing time. This decreased crystallinity reaffirms the DSC observation concerning the slight interface increase. Moreover, X-ray measurements were performed on the original PEEK and PEI films, revealing that narrower peaks of PEEK XRD curves manufactured at 350 ${}^{{}^{\circ}}$C are indicative that PEEK melting had been achieved during processing and that the slower cooling rate led to a higher crystallinity than pristine PEEK.

Lastly, thermomechanical properties were measured by DMA in cantilever mode, from −140 ${}^{{}^{\circ}}$C to 300 ${}^{{}^{\circ}}$C for the two samples processed at 350 ${}^{{}^{\circ}}$C. Figure 6 shows the profiles of both samples and their comparison with pure PEEK and PEI. The most relevant observation is that the processing time has no significant influence, since both samples displayed a similar behavior. The two glass transitions appear isolated, slightly higher in the case of PEEK and lower for the PEI layers. Again, as expected after the DSC analysis, the processing time affected these shifts, which were more pronounced for the samples processed for 120 min. Then, the limited interdiffusion of the PEEK/PEI layers was confirmed even after a long processing time. As this is a temperature-driven process, the interpenetration of polymer chains between these layers was much slower than that in our previous work at higher processing temperatures [33]. These results are in contrast to those obtained at around 400 ${}^{{}^{\circ}}$C by other authors, where only one glass transition was observed in DMA tests due to the high degree of mixture induced on the PEEK/PEI blends at this processing temperature [12,40].

Nevertheless, even without a significant decrease in crystallinity, this slight interface growth due to the extended processing time could affect the mechanical behavior of the samples, mainly when subjected to bending (as these experiments were performed). Under these conditions, a slightly higher storage modulus in the sample processed for 120 min could be expected. Results from the mechanical tests are presented in the next section.

### 3.3. Mechanical Properties of PEEK and PEI

Representative engineering stress versus strain responses of laminate samples of neat and multilayer polymers are presented in Figure 7a,b, respectively.

Neat polymers, as can be seen in Figure 7a, exhibit a damage-tolerant behavior beyond their ultimate stress points $\sigma_{UTS}$. While PEI undergoes a first stress drop followed by a plateau before failing, PEEK displays a gradual decrease in its load-bearing capacity, thus achieving a greater strain to failure. As shown in Figure 7b, multilayered PEEK/PEI blends present a similar response, being capable of absorbing a similar amount of energy as compared to pure PEEK, independently of consolidation time. In fact, the curves corresponding to specimens manufactured at 350 ${}^{{}^{\circ}}$C for 10 and 120 min essentially coincide.

For a better understanding of the tensile behavior, a close examination of the strain fields in the specimens after necking was performed. Figure 8 shows the contour of axial $\varepsilon_{yy}$ and transverse $\varepsilon_{xx}$ strains measured with image correlation just before the specimen failure.

The strain fields in the necking zone of the specimen show that the neat PEEK polymer can reach axial strain values $\varepsilon_{yy}$ close to 63%, whereas neat PEI does not exceed 44%. In this regard, both multilayers can reach intermediate values: a maximum axial strain close to 52% and 59% for PEEK/PEI@350t10 and PEEK/PEI@350t120, respectively. The transverse strain $\varepsilon_{xx}$ field shows that all the polymers studied have maximum transversal strains of around 17%, with this value being present in smaller areas of neat PEI as compared to the other tested specimens. Again, no significant differences were found between the multilayers consolidated for 10 and 120 min.

The elastic modulus (*E*), ultimate tensile strength ($\sigma_{uts}$), and strain at break ($\varepsilon_{b}$) derived from the stress–strain curves are provided in Figure 9a–c.

In terms of the mechanical properties of the manufactured polymers, PEEK, as expected, stands out with its values for the elastic modulus *E* and strain to failure $\varepsilon_{b}$ as compared to the PEI and PEEK/PEI multilayers. Its ultimate stress strength $\sigma_{UTS}$ lies slightly over the neat PEI and PEEK/PEI multilayered blends. The multilayers exhibit a good combination of the mechanical properties of both neat, independently of the processing time. The elastic modulus of the PEEK/PEI MLP can be approximated by the rule of mixtures, with a global modulus close to the arithmetic means of both neat PEEK and PEI, which is expected from a volume proportion of 50/50 because we are able to retain the crystallinity of PEEK. This is essentially the best mechanical performance attainable, in contrast to the properties of the extruded blends. Other research groups have previously studied the tensile properties of PEEK/PEI blends with the same volume ratio, but manufactured by means of extrusion processes, which led to a complete mixing of both polymers. The value reported for the elastic modulus fell well below the one predicted by the rule of mixtures [41]. On the other hand, the strain at break of PEEK/PEI MLP was closer to the neat PEEK strain to failure values than PEI, suggesting that despite being processed in equal proportions, failure mechanisms are dominated by the PEEK fraction.

### 3.4. Nanoindentation across Transversal Section of PEEK/PEI MLPs

The results of nanoindentation tests for PEEK/PEI multilayers are presented in Figure 10. The reduced elastic modulus $E_{r}$ was measured along the region of interest (ROI) indentation distance *x* (see Figure 3), excluding the values that were ±25 μm from the PEEK/PEI interface.

Figure 10a,b show that the reduced modulus of PEEK drops significantly in the vicinity of the interface with PEI, as shown between the dotted line and the PEI phase in Figure 10, while the values of PEI $E_{r}$ remain constant throughout its thickness. This fact is related to the governing mechanisms of the interfacial diffusion of PEEK/PEI; PEI chains, which have greater mobility, penetrate into PEEK, consequently hindering compact molecular packing and reducing its crystallinity, as previously reported in [42]. PEI remains amorphous.

Nanoindentation serves as a tool for assessing the effect of consolidation time at 350 ${}^{{}^{\circ}}$C: First, as processing time increases, PEI has more time to diffuse into PEEK, thus reaching a longer distance and increasing the thickness of the homogeneous interfacial region. This can be seen in Figure 8b, where the typical values for the PEEK reduced modulus are located in a smaller area. This higher interpenetration also corroborates the results from DSC, DMA, and X-Ray diffraction, where PEEK crystallization is more clearly inhibited under a longer processing time, and where glass transitions of both polymers were drawn closer to each other due to the growing interface. In addition to this, it seems that after a long processing time, the reduced moduli of both PEEK and PEI are globally affected as there is a slight decrease in their values. Nevertheless, unless consolidation time exceeds two hours, mechanical properties are not expected to be significantly affected.

## 4. Conclusions

In order to obtain good mechanical properties, the processing technique has to ensure that PEEK maintains a certain degree of crystallinity and that both polymers benefit from their excellent miscibility and exhibit good adhesion by means of an interfacial homogeneous zone. In this work, we studied the processing parameters (consolidation time and temperature) that optimize the mechanical properties of PEEK/PEI multilayer blends when a 50/50 volume ratio is used. The main findings of this study are listed below:Consolidation temperature has to exceed the melting point of PEEK (340 ${}^{{}^{\circ}}$C) so that both polymers are in a viscous state and the molecular chains are mobile enough to achieve interpenetration.350 ${}^{{}^{\circ}}$C seems to be the optimal temperature. On the one hand, it is sufficient to guarantee good adhesion for a consolidation time as short as 10 min. On the other hand, the diffusivity of PEI at this temperature is still relatively low.This slow diffusion kinetics entails that under standard industrial consolidation times, the interfacial region that contains the homogeneous blend does not grow excessively and is confined to a thin layer.Outside the interfacial region, both polymers remain separate phases. PEEK is able to preserve a high degree of crystallinity, and the mechanical response can be described by the rule of mixtures.

Lastly, some general considerations can also be made: As our results show, multilayered PEEK/PEI blends are a feasible alternative in which both polymers are combined in order to obtain a material with good mechanical properties. These multilayers are also expected to display other comparative advantages, such as improved thermal stability or, depending which polymer is selected for the outer surface, good tribological behavior or thermo-oxidative stability. Depending on the particular configuration and design constraints, they could even be welded to other components made of PEEK or PEI without any adhesive agent. Such joints should be easy to repair through the controlled application of heat (e.g., through laser or ultrasonic welding). These assets, together with their reduced cost as compared with neat PEEK alone, make PEEK/PEI multilayers a promising material for automotive, marine, or aerospace applications.

## Figures and Tables

**Figure 1 polymers-14-04597-f001:**
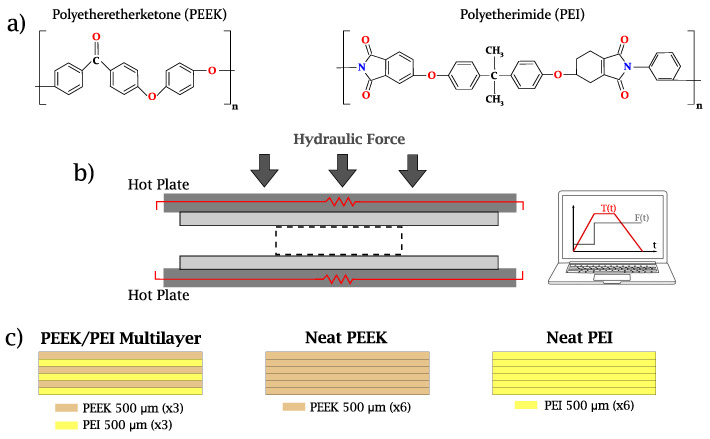
(**a**) Molecular structure of PEEK and PEI. (**b**) Schematic representation of the processing of multilayer PEEK/PEI laminates by means of hot-press molding. (**c**) Stacking sequences for the multilayer polymer blend and neat polymers.

**Figure 2 polymers-14-04597-f002:**
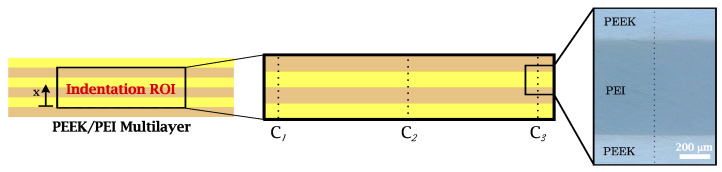
Schematic representation of a nanoindentation grid on the transversal section of the PEEK/PEI blends.

**Figure 3 polymers-14-04597-f003:**
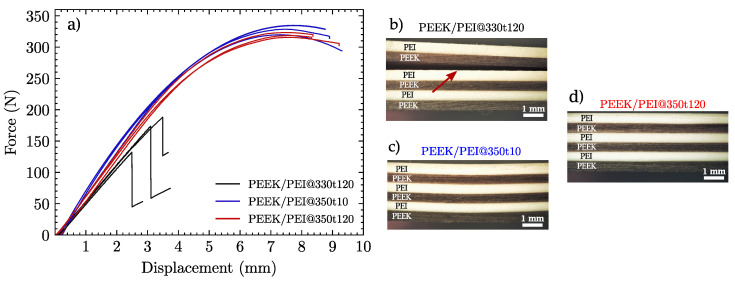
(**a**) Representative load–displacement curves recorded during the three-point bending test of the multilayers. (**b**–**d**) Macrographs of postmortem specimens: (**b**) PEEK/PEI@330t120, (**c**) PEEK/PEI@350t10, and (**d**) PEEK/PEI@350t120. The red arrow in (**b**) marks the position of a delamination crack.

**Figure 4 polymers-14-04597-f004:**
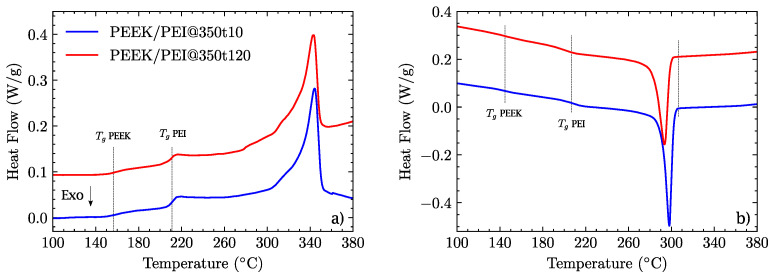
(**a**) DSC melting curve for the PEEK/PEI multilayer processed at 350 ${}^{{}^{\circ}}$C for 10 and 120 min. (**b**) DSC cooling ramp after melting.

**Figure 5 polymers-14-04597-f005:**
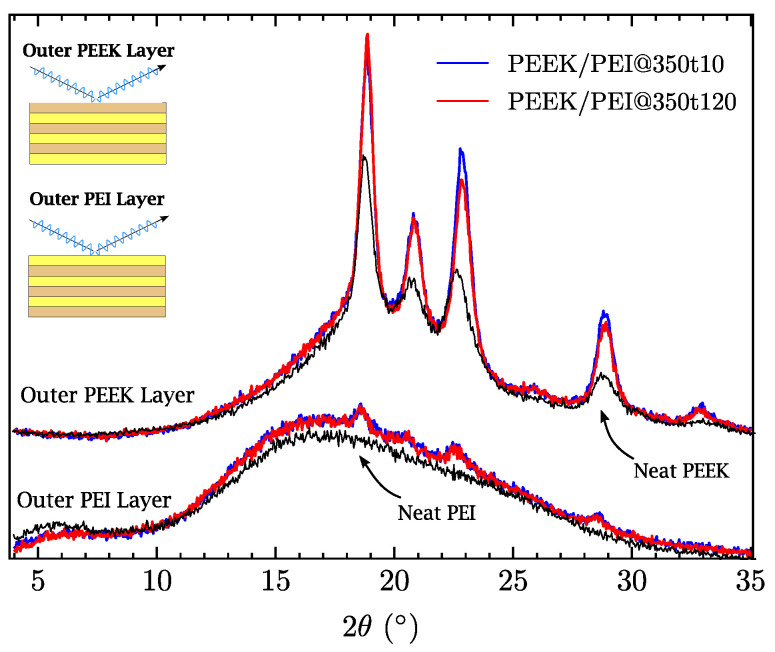
Diffractograms of as-received PEEK–PEI films and multilayer PEEK/PEI polymers processed at 350 ${}^{{}^{\circ}}$C for 10 and 120 min. Upper curves: Diffractograms taken from the reflection of the outer PEEK layer; Lower curves: Diffractograms taken from the reflection of the outer PEI layer.

**Figure 6 polymers-14-04597-f006:**
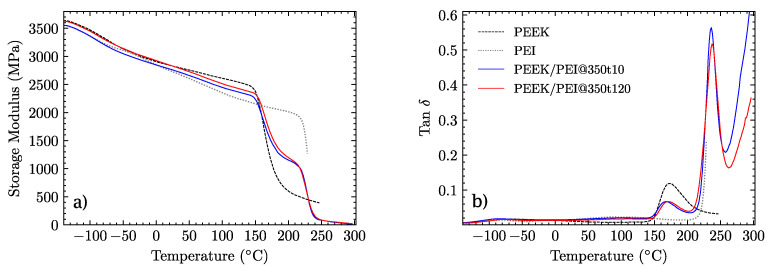
DMA plots of pristine PEEK and PEI and multilayer PEEK/PEI processed at 350 ${}^{{}^{\circ}}$ C for 10 and 120 min. (**a**) Storage modulus. (**b**) Ratio of the loss modulus to the storage modulus (tan δ).

**Figure 7 polymers-14-04597-f007:**
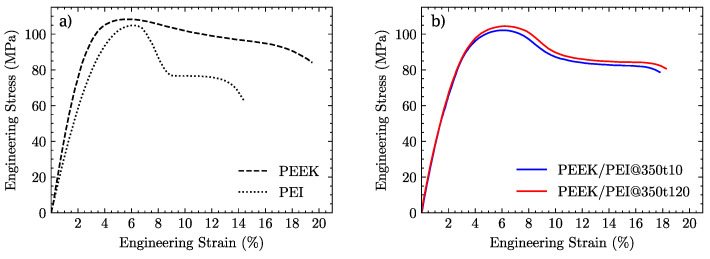
Stress–strain curves of: (**a**) neat PEEK and PEI polymers and (**b**) PEEK/PEI MLPs processed at 350 ${}^{{}^{\circ}}$C for 10 and 120 min.

**Figure 8 polymers-14-04597-f008:**
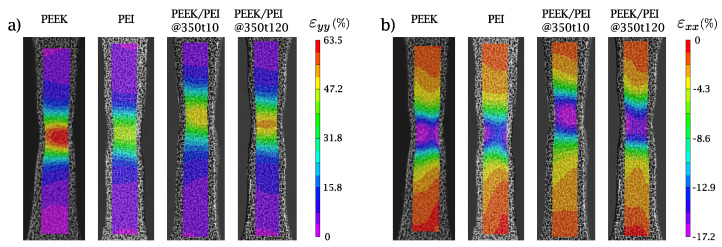
Visualization of the engineering strains (**a**) $\varepsilon_{yy}$ and (**b**) $\varepsilon_{xx}$ of the different polymers manufactured.

**Figure 9 polymers-14-04597-f009:**
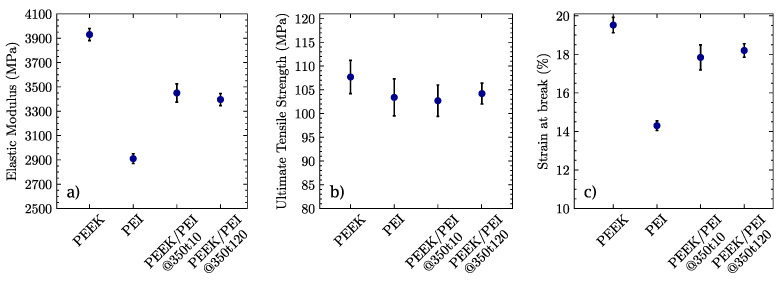
(**a**) Elastic modulus, (**b**) tensile strength, and (**c**) elongation at break of neat PEEK and PEI polymers, and PEEK/PEI MLPs processed at 350 ${}^{{}^{\circ}}$C for 10 and 120 min.

**Figure 10 polymers-14-04597-f010:**
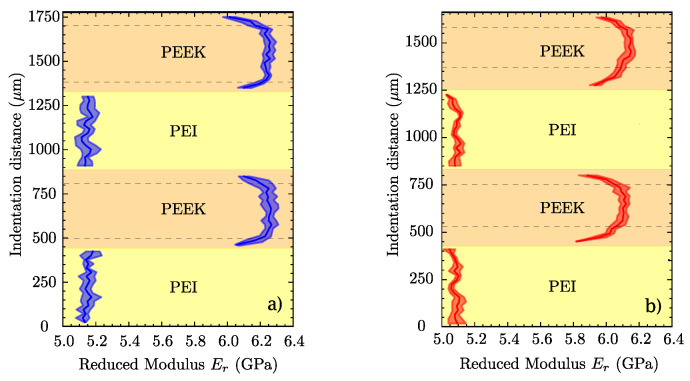
Profiles of average reduced modulus $E_{r}$ for (**a**) PEEK/PEI@350t10 and (**b**) PEEK/PEI@350t120.

## Data Availability

Not applicable.

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
