# Peer review of "Optimization of Processing Conditions and Mechanical Properties for PEEK/PEI Multilayered Blends"

_polymers, 2022, doi:10.3390/polym14214597_

Round 1
Reviewer 1 Report
The authors used PEEK/PEI multilayers with three fabrication conditions. The layers adhesion for samples fabricated below the PEEK melting points showed poor adhesion, thermal, and mechanical properties. The research idea is very unique with outstanding results interpretation. However, I have a few points for the author's consideration.
1- Have you done any Microscopic imaging of the sample cross-section to calculate the exact layer thickness, defect formations, and adhesion? For example, the layer thickness could be not equal due to irregular pressure distribution or fabrication conditions.
2- In figure 3, it has been mentioned there is delamination in the sample which is heated to 330 oC. This has been pointed out by the red arrow on the upper right picture of the macrograph of the specimens after bending tests. Is this happening for all five samples or for only just one of them?
3- For the XRD curve, could you please include the control samples, specifically before lamination this could indicate the structural changes during the fabrication process.
Author Response
Please, see the attached pdf document.

Reviewer 2 Report
Even though the topic and idea are not very novel, the whole work is designed properly, experimental data support their conslusion, and the multilayer structure reported here also provides new guidelines for polymer blend design. The writing is clear as well. I support the publication of this work.
Author Response
Even though the topic and idea are not very novel, the whole work is designed properly, experimental data support their conclusion, and the multilayer structure reported here also provides new guidelines for polymer blend design. The writing is clear as well. I support the publication of this work.
Authors appreciate the comments by the reviewer.
Reviewer 3 Report
1) This is an interesting study, but some additional experimental conditions of the three-point bending test are also needed. In this paper, only tests at 330 °C and 350 °C are conducted, but it is difficult to explain that 350 °C is the optimal process parameter. It is recommended to do several sets of tests and comprehensively considers the stiffness, strength and toughness to find the optimal temperature parameters.
2) It is suggested to evaluate the interfacial shear properties of the multilayer plastics. Maybe multiple components can be directly connected by heating without adhesive joints.
Author Response
The authors are grateful for the careful reading of the manuscript and comments.
1) This is an interesting study, but some additional experimental conditions of the three-point bending test are also needed. In this paper, only tests at 330 °C and 350 °C are conducted, but it is difficult to explain that 350 °C is the optimal process parameter. It is recommended to do several sets of tests and comprehensively considers the stiffness, strength and toughness to find the optimal temperature parameters.
Authors agree with the reviewer that a full, rigorous optimization of the mechanical properties with respect to the processing temperature would require a batch of specimens sequentially manufactured over a range of temperatures followed by three-point bending tests. However, it is also crucial to consider the processing setup's limitations: the hot press can achieve reasonable temperature control over the full multilayer with a precision of a few Celsius if processing times are long but less accurate if they are shorter. Therefore, in order to obtain precise results, the size step temperature of 10 ºC seems reasonable. Regarding the high-temperature regime (35 °C over melting temperature), it was already known from our previous work that a manufacturing temperature of 380º benefitted from polymer miscibility and provided good adhesion, but led to poorer mechanical properties because extensive interdiffusion inhibited PEEK crystallinity. On the lowest end, it was extremely important to determine whether processing below the melting point of PEEK (around 340ºC) produced good adhesion. For this reason, 330 ºC was selected as the first processing temperature. Results prove that the interface is extremely weak. 340 ºC was not considered, because the exact melting temperature of PEEK depends on the particular polymer. At 350 ºC, the interface exhibited good strength when processed for 10 and 120 minutes. Since PEEK crystallinity is preserved and the material works as a composite governed by the rule of mixtures, there are no additional gains to be expected if the temperature increases beyond this point, particularly because diffusion can only impair the mechanical properties. From the viewpoint of production and cost, shorter manufacturing time is always preferred; therefore, 350ºC@10min might be regarded as the optimal combination. To clarify this point, additional information was added in the “Samples preparation” section on page 3, lines 91-95.
2) It is suggested to evaluate the interfacial shear properties of the multilayer plastics. Maybe multiple components can be directly connected by heating without adhesive joints.
We share the opinion by the referee that components can be welded without adhesive agents with dissimilar chemical composition. Since this is a relevant point, we now mention it in the manuscript. However, measuring interfacial properties falls beyond the scope of this work. We wanted to ensure through flexural tests that the interface (of specimens above the melting point of PEEK) was at least as strong as the individual layers. Once this point has been confirmed, measuring the interfacial properties is not straightforward (the crack can propagate along the interface and through the bulk of one of the polymer layers) and specific tests are required. Future research is intended to address this topic.
Reviewer 4 Report
Please refer to the attached file

Author Response
Thank you for your comments. We appreciated that the reviewer provided considerable insight into our manuscript.
ABSTRACT:
- Page 1, Line 1: ‘PEEK/PEI’ should write full name and (PEEK/PEI) in a bracket. Please take note, for 1st time appear in manuscript, should write full name and then write abbreviation/symbol in a bracket. Please check the whole manuscript and do correction accordingly.
Thanks for the comment. We have incorporated all full names as corresponds.
- Page 1, Lines 9-10: ‘DSC, DMA & XRD’ Please take note, for 1st time appear in manuscript, should write full name and then write abbreviation/symbol in a bracket. Please check the whole manuscript and do correction accordingly.
Thanks for the comment. We have incorporated all full names as corresponds.
- At the end of abstract. Please add significant of findings from this study to related industries.
Thanks for the suggestion. Main findings have been added to the abstract.
KEYWORDS:
- Please write as Multilayer polymers (MLPs); Polyetheretherketone (PEEK); Polyetherimide (PEI)
Thanks for the suggestion. This has been corrected.
- Please add another 2 keywords
Thanks for the suggestion. 5 keywords have been added to the original manuscript.
INTRODUCTION:
- Page 2: Line 55, at beginning of paragraph, please highlight the novelty of this study and at the end of paragraph, please state the important of finding to related industry.
Thanks for the suggestion. The corresponding novelty and findings related to the industry have been added to the introduction.
MATERIALS AND METHODS
- Page 3, Figure 1: Please unbold (a), (b) and (c) in figure and caption
Thank you. This has been corrected accordingly.
- Page 3, Line 109: Please state brand, manufacturer and country for X-Ray diffractometer.
These data have been added to the manuscript.
- Page 4, Lines 123-124: Please remove ‘as 123 described in’
Thanks for the comment. This has been corrected.
- Page 4, Lines 132-133: Please change ‘according to the following equation:’ to ‘as expressed in Equation 1 [X]:’ Please add citation for Equation 1 accordingly
Thank you for your comment. This has been corrected.
- Page 4, Figure 2: Please improve the quality. Make sure all values/wordings are readable. Please use standard size and font type for all wordings and values at all figure. Suggest to use Palatino Linotype (type font), text size 9. Please check the whole manuscript and do correction accordingly.
Thanks for the suggestion. The quality of figure 2 has been improved.
RESULTS AND DISCUSSION
- Page 5, Figure 3: Should label (a), (b), (c) and (d). Please add label the red arrow in figure
This has been corrected.
- Page 5, Lines 159-176: Please compared results obtain with previous researchers (other researchers beside of your previous works). Is it inline? or contradict? And please discuss critically with citation/s why it happens
Thanks for the comment. We have added some references that report similar results and have been compared with our current work. See new manuscript, page 6, lines 196-198.
- Page 5, Line 173: Should write as ‘…our previous investigations [33].’
This has been properly corrected.
- Page 5, Figure 4: Please label (a) and (b) for Figure 4
Thanks for the comment. Corresponding labels were incorporated in Figure 4.
- Page 5, Figure 4: Please remove (left) and (Right) in the caption and please mention (a) and (b)
Thanks for the comment. These observations have been made.
- Page 6, Lines 177-200: Please compared results obtain with previous researchers (other researchers beside of your previous works). Is it inline? or contradict? And please discuss critically with citation/s why it happens
Thanks for the comment. We have incorporated studies that report similar results, which have been compared with our current work. See new manuscript, page 7, lines 222-225.
- Page 6, Figure 5: Please improve the quality. Make sure all values/wordings are readable. Please use standard size and font type for all wordings and values at all figure. Suggest to use Palatino Linotype (type font), text size 9. Please check the whole manuscript and do correction accordingly.
Thanks for the suggestion. The quality of figure 5 has been improved.
- Page 6, Figure 6: Please label (a) and (b) for Figure 4
This has been properly corrected.
- Page 6, Figure 6: Please remove (left) and (Right) in the caption and please mention (a) and (b)
This has been properly corrected.
- Page 7, Caption Figure 7: Please unbold (a) and (b). Please check the whole manuscript and do correction accordingly.
This has been properly corrected.
- Page 7, Caption Figure 8: Please unbold (a) and (b). Please check the whole manuscript and do correction accordingly
This has been properly corrected.
- Page 7, Figure 8: Please improve the quality. Make sure all values/wordings are readable. Please use standard size and font type for all wordings and values at all figure. Suggest to use Palatino Linotype (type font), text size 9. Please check the whole manuscript and do correction accordingly.
Thanks for the suggestion. The quality of figure 8 has been improved.
- Pages 7-8, Lines 204-233: Please compared results obtain with previous researchers. Is it inline? or contradict? And please discuss critically with citation/s why it happens
Thanks for the comment. We have discussed previous literature references reporting on the tensile behavior of PEEK/PEI blends. See new manuscript, page 9, lines 261-266.
- Page 8, Caption Figure 9: Please unbold (a), (b) and (c). Please check the whole manuscript and do correction accordingly.
Thank you. This has been properly corrected.
- Page 8, Caption Figure 10: Please unbold (a) and (b). Please check the whole manuscript and do correction accordingly
Thank you. This has been properly corrected.
CONCLUSIONS:
- Page 9, Lines 257-275: Please numbering the findings using point i, ii, iii etc
Thanks for the suggestion. Conclusions have been rewritten, and the main findings are correspondingly listed with bullet points.
- Page 9, Lines 257-275: At last, in 1 paragraph, highlight the significant of findings for related industries and suggest for future works.
Thanks for this constructive suggestion. Some comments on the significance of findings and future work related to the topics covered in our manuscript have been included.
Round 2
Reviewer 3 Report
-